# Effect of the Complexity of the Customs Tax System on the Tax Effort

**Jazmín González Aguirre [1,\*] and Alberto Del Villar [2,\*]**

1   Economics and Business Management, University of Alcala (UAH), 28802 Alcala de Henares, Spain
2   Department of Applied Economics, University of Alcala (UAH), 28802 Alcala de Henares, Spain
\*   Correspondence: jazmin.gonzalez@edu.uah.es (J.G.A.); alberto.delvillar@uah.es (A.D.V.)

**Abstract:** This paper empirically analyses the effects of the tax complexity and other elements, such as natural resource revenues, public expenditure, and the capacity of the statistical system, on the efficiency of Ecuadorian Customs Administration. For this purpose, the methodology used consists of modeling a stochastic production frontier whose estimation procedure is based on the maximum likelihood method for a data panel composed of six countries: Bolivia, Chile, Colombia, Ecuador, Peru, and Panama for the period from 2006 to 2017. The results of the study show that the countries whose tax system has a lower degree of complexity present a better level of revenue collected and tax effort as well as an improvement in the quality and dissemination of national statistical data. Furthermore, this paper provides evidence that the tax effort tends to decrease when the price of crude oil is on the rise.

**Keywords:** tax complexity; tax effort; efficiency; customs; international trade; Ecuador; Latin America

## 1. Introduction

Tax collection is a matter of concern to public managers. There is an acceptable tax level that allows one to achieve a high level of tax collection and does not discourage economic activity. Setting tax rates in such a way as to allow the maximum tax collection and the minimum loss of economic activity is something that almost all public managers pursue. In this sense, the effect that tax policy has on a country's economic development is crucial, and it is not free from controversy.

In Latin America and the Caribbean, the economic development strategy has been strongly linked to international trade with the application of a series of trade policies considered fundamental in order to compete in a globalized context. The number of treaties and trade agreements confirms this; for example, Chile is the country with the highest number of trade treaties in the region, followed by Peru and Mexico (WTO 2019).

Similarly, customs tariffs play an important role as a tax policy tool. On the one hand, customs taxes are a significant source of public revenue, as is the case for Value-Added Tax (VAT), which is generally the main source of revenue for governments in the region. Furthermore, customs tariffs run the risk of being used as a tool to protect domestic production or equilibrate the balance of payments, especially when investment policies are not effective.

The above is exemplified by the case of Ecuador (8.19%), a country that had one of the highest (restrictive) tariff rates in the region in 2019, only surpassed by Belize (11.17%), Venezuela (10.19%), and Cuba (8.54%) (World Bank 2019, Tariff rate, applied, weighted mean, all products).

After 2007, Ecuador implemented a selective import substitution policy with the objective of strengthening domestic production and slowing down the trade deficit. Prior to 2007, the industrial sector's share was less than 13% of the GDP, so the government

aimed to reach a minimum of 25% of the GDP by 2017. In addition, the Government of Ecuador decided to use a tariff policy as an instrument to level the balance of payments in order to maintain the dollarization monetary process in the country's economy. To support the economic policy of selective import substitution, it was necessary to adopt a range of tariff and non-tariff measures.

Regarding tariff barriers, the country applies two types of tariff duties: ad valorem and compound tariffs. The latter apply primarily to textiles, footwear, ceramic plates and tiles, alcoholic beverages, televisions, and motor vehicles. The tariff levels faced by certain goods are high; for example, garments and other textile items are charged tariffs ranging from 178.9% to 422.2% considering their ad valorem equivalent (EVA). Between 2007 and 2017, the Most-Favored Nation (MFN) clause tariff applied (simple average) went from 11.7% to 12.2% (all products).

In 2009, 2015, and 2016, the country adopted a balance-of-payments safeguard, and approximately 40% of imports experienced tariff surcharges. The level of the tariff surcharge varied between 5% and 45%, with the 25% and 45% surcharges taxing 58.6% of the universe of products subject to the safeguard measure. For the period 2011–2018, the MFN tariff applied had already exceeded the bound tariff by 28% of the total set of tariff lines (WTO 2019). All these measures resulted in a significant reduction in imports, which, however, was insufficient to prevent the deficit from increasing.

Within the period 2007–2017, a number of changes were introduced into the regulatory framework governing the importation of certain products, such as cotton, clothing, and fruits and vegetables. These measures included the use of an importer's registry, a certificate of recognition, the granting of automatic and non-automatic import licenses, certificates of recognition for products subject to technical regulations, as well as import quotas for motor vehicles and cell phones.

In the field of the revenues collected by the customs service, the question focuses on determining the level of the external tariff that allows for the maximum tax collection but does not limit the production of goods and services. In other words, the tax effort made by companies and citizens must be neutral in relation to their economic activities. To determine this level, a three-element analysis is required.

First, the tax capacity, defined as the maximum level of tax revenue that a country can achieve, is determined. For this purpose, certain factors affecting the tax capacity are considered, such as the degree of productivity and economic development, inequalities in the income distribution, trade openness, education, and inefficiencies in tax collection. For the construction of the stochastic tax frontier, we followed the study of Pessino and Fenochietto (2013). However, some variables were replaced by the authors in order to approximate the customs context.

Second, the tax effort, defined as the ratio between the actual revenue and the tax capacity, is determined. The tax effort represents the inability of the tax administration to collect the maximum amount of revenue possible.

Finally, the degree to which the complexity of a tariff system affects the tax effort as well as the incidence of other factors, such as natural resource rents, public expenditure, and statistical capacity, are determined.

This paper uses the econometric model of the "stochastic tax frontier" for a panel of data to determine these elements. We followed the study by Pessino and Fenochietto (2013) and other studies that have used the stochastic frontier model to estimate the tax effort (Jha et al. 1999; Esteller 2003; Alfirman 2003; Cyan et al. 2013). However, they all focus on the domestic tax system, and do not include an analysis that exclusively addresses the issue of the customs tax system. This issue still has a degree of importance in developing countries, where the domestic tariff structure and tax rate levels are used as instruments to balance trade and/or an important source of financing for public expenditure.

This paper is structured as follows. Section 2 presents a brief review of the relevant literature. The authors draw on Ecuador's experience to identify what factors may have had a negative impact on the customs tax effort. Section 3 explains the set of variables used

to estimate the stochastic tax frontier as well as the variables used to estimate the effect on the tax effort. Section 4 analyses the most significant results. Finally, Section 5 presents the conclusions of the empirical study.

## 2. Definitions: Tax Effort, Revenue Potential, and Tax Complexity

One of the first approaches to the definition of 'tax effort' was that of Frank (1959), who defined it as the ratio between total revenue as a share of GDP and a country's per capita income, which is currently known as the absolute tax effort. However, this ratio presents some difficulties when making comparisons between countries, especially when they have different tax structures, economic conditions, and demographics. To solve this problem, most works on tax effort focus on using the concept of relative tax effort, which is defined as the ratio of a country's real tax revenue to its tax capacity (Minh Le et al. 2012; Khwaja and Iyer 2014). In other words, relative tax effort is an indication of how close or distant a country is to its maximum revenue-raising capacity.

Tax capacity or revenue potential refers to the ability of an economy to generate a certain level of taxation as a function of various explanatory variables and can be easily estimated using stochastic frontier analysis (SFA) as we will see below.

According to Budak and James (2018), in the economic literature there is no definition for tax complexity; instead, authors prefer to describe various factors that contribute to its analysis, such as Smith's (1776) criteria: fairness, certainty, efficiency, and simplicity. We will consider that lowering the tax complexity means reducing the number of taxes and leaving only those taxes that are the easiest to pay and collect, as defined by Thuronyi (1996). Therefore, throughout this paper, tax complexity will be related to the characteristics of the tax system, such as the number of taxes and the goods that are taxed.

## 3. Literature Review

As far as we are aware, empirical papers studying the degree of complexity of the customs tax system are scarce. Tran-Nam (2004) points out that, although the simplicity of the tax system was discussed in the early days of modern economics by famous figures such as Smith (1776), this issue has been ignored over time by scholars of public finance economics, resulting in a lack of a theoretical framework with which to assess the impact of tax simplification in tax reform.

In order to discuss whether or not the complexity of the tariff system influences the tax effort, we identified the following variables as being of interest.

- Construction of the Stochastic Tax Frontier

The first studies on the stochastic frontier model (SFA) were conducted by Aigner et al. (1977) and Meeusen and van Den Broeck (1977), who econometrically modeled a production function to estimate technical efficiency parametrically using cross-sectional data.

Battese and Coelli (1992, 1995) extended the model proposed by the aforementioned authors to estimate the parameters of the stochastic frontier and technical inefficiency simultaneously for a panel of data. This technique has the advantage of being able to estimate the technical change in both the stochastic frontier and the time-varying technical inefficiency. On the other hand, the use of an SFA model with a panel of data minimizes the problems caused by multicollinearity and omitted variables (Hsiao 1986).

Based on this estimation methodology, stochastic frontier models have commonly been used to estimate the degree of tax effort (Jha et al. 1999; Maekawa and Atoda 2001; Esteller 2003; Alfirman 2003; Pessino and Fenochietto 2013; Cyan et al. 2013), taking into account the following considerations.

We employed the stochastic tax frontier analysis based on a panel of data to estimate the tax effort, which is measured by relating the effective revenue to the potential revenue or tax capacity. The actual revenue collection is the actual amount of revenue obtained after the payment of taxes, while the potential revenue is an unobservable variable, i.e., it cannot be controlled by the tax administration, and refers to the maximum amount of tax revenue

that is possible to collect given the economic characteristics of a country or region (Khwaja and Iyer 2014).

Mathematically, the potential revenue or tax capacity would be given by the Production Possibilities Frontier (PPF), which is constructed through the graphical representation of the maximum amounts of revenue that in a specific time can be obtained through the resources available. So, the stochastic tax frontier for a panel of data can be expressed mathematically as follows:

$$Y_{it} = \exp\left(x_{it}\,\beta + v_{it} - \mu_{it}\right) \tag{1}$$

where:

$I$ represents the number of observations;

$t$ is a time period;

$y_{it}$ is the actual revenue collected by the Customs Administration ($i$) in year ($t$);

$x_{it}$ corresponds to the set of variables that explain the potential revenue of $i$ in year ($t$);

$\beta_0$ is a constant;

$\beta$ represents the input parameters of the production function;

$v_{it}$ is a random error and an independent and identically distributed stochastic component with a mean of zero and a constant variance $N\left(0, \sigma_v^2\right)$ *i.i.d* that represents any exogenous factor that cannot be controlled by the Customs Administration, e.g., a tax exemption that affects revenue collection. It may also refer to measurement errors. $v_{it}$ may take a positive or negative value;

$\mu_{it}$ is a random stochastic component of the technical inefficiency and a non-negative term that is assumed to be independently distributed. In this context, the inefficiency represents the inability to achieve the maximum amount of revenue collection. The components $v_{it}$ and $\mu_{it}$ are assumed to be independent of each other and the estimators.

The socio-economic variables that determine the potential revenue collection are reported in the study by Pessino and Fenochietto (2013), these being: the degree of productivity and economic development, income distribution, trade openness, education, and inefficiency of tax collection. Pessino and Fenochietto (2013) use as a measure of tax collection inefficiency the share of agriculture in GDP, assuming that it is more difficult to exercise taxation control over some economic sectors such as agriculture due to the small number of farmers. However, in this paper the size of the shadow economy is used as a measure of the inefficiency of tax collection because it more accurately estimates the size of the economic activity that escapes tax control.

- Factors influencing the Tax Effort

According to the model of Battese and Coelli (1995), the inefficiency term $\mu_{it}$ is expressed as a function of explanatory variables, a vector of unknown coefficients, and a random error. Therefore, this function could be represented as:

$$\mu_{it} = \delta\,Z_{it} + W_{it} \tag{2}$$

In this equation:

$Z_{it}$ is the set of exogenous variables that would explain the inefficiency in tax collection;

$\delta$ is a vector of coefficients to be estimated; and

$W_{it}$ is a random variable defined as a truncated normal distribution with mean of zero and a constant variance. The point of truncation is $\delta\,Z_{it}$, so that $W_{it} \geq -\delta Z_{it}$.

Equations (1) and (2) are estimated by the maximum likelihood (MLE) method in the simultaneous equations model. Consequently, the tax effort defined as the ratio between the actual revenue and the tax capacity can be obtained by the following expression:

$$\text{Tax Effort} = \frac{Y_{it}}{\exp\left(x_{it}\beta + v_{it}\right)} = \frac{\exp\left(x_{it}\beta + v_{it} - \mu_{it}\right)}{\exp\left(x_{it}\beta + v_{it}\right)} = \exp\left(-\mu_{it}\right) = \exp\left(-\delta\,Z_{it} - W_{it}\right) \tag{3}$$

Then, $\mu_{it}$ is obtained by truncation of the normal distribution with mean $\delta\,Z_{it}$ and variance $\sigma^2$ with a value on a scale of 0 to 1.

## 4. Methodology and Data

Following the strategy of Pessino and Fenochietto (2013), the efficiency of the Customs Administration was determined in the form of the tax effort, which was measured as the relationship between the actual tax collection and the tax capacity. Pessino and Fenochietto use as an econometric approach the methodology proposed by Battese and Coelli (1995), which is based on a stochastic production frontier fitted by balanced panel data and estimated using the maximum likelihood (MLE) method. This model has the advantage of allowing us to:

- Estimate the tax capacity or potential tax revenues, in which the maximum level of tax revenue considered is influenced by the socio-economic characteristics of the country; and
- Decompose the error term into two components (random noise and the tax effort), so it is possible to model the tax effort through a set of variables. For example, we can determine whether the complexity of the customs tax system has any influence on the revenue collection efficiency.

### 4.1. Model Specification

To estimate this model, we used a balanced data panel for a 12-year period (2006–2017) comprising six countries: Bolivia, Chile, Colombia, Ecuador, Peru, and Panama. Table 1 provides a description of all variables.

The variables that determine the tax revenue potential are defined as follows:

$$ln(R\_pib)_{it} = \ \beta_0 + \beta_1 ln(PIB\ pc)_{it} + \beta_2\ ln(PIB\ pc)_{it}^2 + \beta_3\ ln(C\_pib)_{it} + \beta_4 ln(Gini)_{it} + \beta_5 ln(Edu\_pib)_{it} \\ + \beta_6\ ln(e\_sum\_pib)_{it} + v_{it} - \mu_{it} \tag{4}$$

The effects of inefficiency are defined as follows:

$$\mu_{it} = \delta_0 + \delta_1(IHH)_{it} + \delta_2(min\_pib)_{it} + \delta_3\ (oil\_pib)_{it} + \delta_4(stat)_{it} + \delta_5\ (gp\_pib)_{it} + \delta_6 Yeardum \tag{5}$$

**Table 1.** Description of Variables.

| Variable | Description |
|---|---|
| | Stochastic Tax Frontier |
| $i$ | Country of Customs Administration |
| $t$ | Year t (t = 2006 … 2017) |
| $\beta_0$ | Constant |
| $\beta_1 \ldots \beta_6$ | Coefficients of elasticities measuring the percentage change in the dependent variable with respect to the unit percentage change in the independent variable |
| $ln(R\_pib)_{it}$ | Logarithm of customs revenue as a proportion of the gross domestic product (GDP) of country *i* in year *t* |
| $ln(PIB\ pc)_{it}$ | Logarithm of the gross domestic product per capita by purchasing power parity (constant 2011 international dollars) for country *i* in year *t* |
| $ln(PIB\ pc)_{it}^2$ | Ln (PIB pc) squared |
| $ln(C\_pib)_{it}$ | Logarithm of merchandise trade as a proportion of the GDP for country *i* in year *t* |
| $ln(Gini)_{it}$ | Logarithm of the Gini coefficient |
| $ln(Edu\_pib)_{it}$ | Logarithm of government expenditure on education as a percentage of the GDP for country *i* in year *t* |
| $ln(e\_sum\_pib)_{it}$ | Logarithm of the shadow economy as a proportion of the GDP for country *i* in year *t* |

**Table 1.** *Cont.*

| Variable | Description |
| --- | --- |
| $v_{it}$ | Assumed to be $N(, \sigma_v^2)$ *i.i.d* error terms independent of $\mu_{it}$ |
| $\mu_{it}$ | The term of inefficiency for country *i* in year *t*. It is a non-negative disturbance term and is assumed to be $N(\mu, \sigma_u^2)$ *i.i.d*. We assumed a half-normal distribution and estimated this term following the procedure suggested by Battese and Coelli (1988), which the SAS software identifies as TE1. |
| | The effects of inefficiency |
| $\delta_0$ | Constant |
| $\delta_1 \dots \delta_6$ | Elasticity coefficients measuring the percentage change in the dependent variable with respect to the unit percentage change in the tax effort |
| $(IHH)_{it}$ | Hirschman–Herfindah concentration index used as a proxy for the degree of complexity of the customs tax system for country *i* in year *t* |
| $(min\_pib)_{it}$ | Mineral rents as a percentage of the GDP for country *i* in year *t* |
| $(oil\_pib)_{it}$ | Oil rents as a percentage of the GDP for country *i* in year *t* |
| $(stat)_{it}$ | Statistical capacity indicator for country *i* in year *t* |
| $(gp\_pib)_{it}$ | Public expenditure as a percentage of the GDP |
| *Yeardum* | Dummy variable for the time period (1 = 2006 … 12 = 2017) |

*4.2. Data Description*

Details of data sources can be found in Appendix A. Data were mainly sourced from annual reports by the Customs Administrations and the Central National Bank. The description of all variables used to estimate the stochastic tax frontier is as follows.

$ln(R\_pib)_{it}$ is the dependent variable formed as a ratio between the customs revenue (R) at the current prices and the nominal GDP. The customs revenue posted on the statistics website of each Customs Administration was used, while the information on GDP was obtained from the national statistics offices.

In terms of the socioeconomic variables that explain the tax revenue potential, we used the following.

$ln(PIB\ pc)_{it}$, the GDP per capita by purchasing power parity (constant 2011 international dollars) is an independent variable commonly used to explain a country's tax capacity taking into account the degree of its economic activity. A directly proportional relationship is expected between the customs revenue and the income levels of the inhabitants. The square of this variable was used in order to capture a potential nonlinear relationship between customs revenue and GDP per capita, $ln(PIB\ pc)_{it}^2$. So, we expect that a high income level of the inhabitants will act in favor of increased customs revenue. This information is available on the United Nations Statistics Division (UNSD) website.

Another economic variable used to estimate the tax revenue potential is the degree of participation in international trade in the economy of a country $(ln(C\_pib)_{it})$, which was calculated as the ratio between the sum of merchandise exports and imports and the GDP value, all in current United States dollars. It was expected that an increased trade flow would have a positive impact on an increase in customs revenue.

The Gini index, $ln(Gini)_{it}$, a coefficient that measures the degree of income inequality between a country's inhabitants, was used under the following assumption: A Gini index close to 1 would indicate high levels of wage inequality, which increase the propensity to form an informal economy and reduce the possibility of obtaining higher levels of income. Therefore, the sign of the Gini index estimator was expected to be negative. The information can be accessed at the data.worldbank.org (accessed on 21 October 2019) website.

Pessino and Fenochietto (2010) estimate tax capacity using the variable $ln(Edu\_pib)_{it}$, defined as government expenditure on education as a percentage of the GDP. However, Cyan et al. (2013) argue that education may not always have a favorable impact on tax revenue because more educational experience may be used to take advantage of mechanisms or legal loopholes for committing tax evasion. Therefore, its impact on customs revenue is undetermined.

In order to estimate the tax capacity, it is also necessary to consider those adverse factors that prevent the level of revenue from increasing. In the case of internal revenue, Pessino and Fenochietto (2010) consider the agricultural value added as a proportion of the GDP, assuming that this sector is more likely to register a higher degree of informality. This is because it is composed mainly of small businesses. In this study, we used the variable shadow economy as a proportion of GDP ($ln(e\_sum\_pib)_{it}$), since it is considered a more precise indicator for estimating the size of economic activity that escapes tax control. The term 'shadow economy' is broad. It includes illegal activities, such as drugs, trafficking in endangered species, and smuggling, and activities of an informal nature such as sales or undeclared work. The information was obtained from a Working Paper written by Schneider 2012 and Schneider and Medina 2018, who estimated of the size of the shadow economy for more than 150 countries between 1991 and 2015. For missing years, we filled in values with the extrapolation method. Therefore, we expected an inverse relationship between the customs revenue and the shadow economy.

The tax effort $exp\,[-\mu_{it}]$ uses values ranging from 0 to 1, with 0 representing a situation of a lower degree of tax effort and 1 expressing exactly the opposite. The variables considered for its explanation are as follows.

The Herfindahl–Hirschman index ranges from 0 to 1; when the value is "0", it indicates the maximum tax complexity, and when the value is equal to "1", it is a sign of the minimum tax complexity. So, the HHI index was expected to have a positive impact on the tax effort in accordance with Jorratt (1996) and Carroll (2009), who, in the case of internal taxes, argue that a highly complex tax system, for example with a multiplicity of levies, exceptions, bans, and tax expenditures, favors an increase in administrative and compliance costs. It makes the auditing process more difficult, increases the time needed to comply with tax formalities, and favors the possibility of evasion.

For the calculation of the HHI index, information on each type of customs tax was used. The standardized HHI index is specified as follows:

$$HHI\ standardized = \frac{\left(\sum_{i=1}^{N} Ri^2 - \frac{1}{n}\right)}{1 - 1/n} \tag{6}$$

where:

$Ri^2$ is the share of customs tax $i$ in the total revenue, squared; and
$n$ is the total number of customs taxes.

This HHI index uses the following types of customs tax, depending on the tax structure of each country: (1) customs duties, such as ad valorem rates, specific rates, or combined rates, and specific types of customs duties, such as antidumping duties, countervailing duties, and tariff-rate quotas (safeguards measure); and (2) other taxes such as value-added taxes (VATs), selective consumption taxes, child development fund taxes, and municipal promotion taxes.

The Herfindahl index is not a perfect measure of the degree of complexity of a tax system, so its interpretation should be performed with caution. For example, this indicator only gives an idea of the number of taxes and the weight of the tax burden on the goods that are taxed. On the other hand, the HHI index does not consider the number of tax exemptions, the number of times a law is modified, the degree of difficulty in understanding the law, the amount of information provided by the tax authorities, the amount of information required to fill out a customs declaration form or apply for a tax return, etc.

Nevertheless, the variable seems to be a feasible measure, as it captures much, but not all, of what it means to define a tax system as simple (Wagner 1976).

In some Latin American countries, the mineral revenue ($(min\_pib)_{it}$) and oil revenue ($(oil\_pib)_{it}$) are important components of the GDP due to exports and fiscal sources. According to Sachs and Warner (1997), a high degree of dependence on natural resources can have a detrimental effect on the national economy. For example, a strong inflow of foreign currency could appreciate the local currency, making non-oil exports less competitive, thus

affecting private investment and production. In addition, the application of certain taxation policies, such as import substitution using tariff and/or non-tariff barriers, is recurrent, with the aim of alleviating the trade deficit in non-oil goods or increasing investment and public spending to boost the national economy.

The World Bank's statistical capacity indicator $((stat)_{it})$ assesses the capacity of the national statistical system. The areas diagnosed are the methodological quality and the source of data as well as the periodicity and timely dissemination of statistical information relevant to public decision-making. The range of the indicator is between 0 (low performance) and 100 (high performance). According to Beccaria (2017), a low performance in this indicator could indicate a lack of incentives to generate tools that allow citizens to evaluate public management. Similarly, Shah (1996) and Weingast (2006) argue that when a government tends to be concerned with increasing tax revenues, the level of accountability and transparency improves, thus discouraging corruption. Therefore, the hypothesis is that the tax effort tends to improve the higher the degree of statistical capacity.

To explore whether total public expenditure as a proportion of GDP has an impact on improving the tax effort, we used the variable $((gp\_pib)_{it})$ as a proxy for institutional spending. A directly proportional relationship between both variables was expected.

*Yeardum* is a dummy variable used to determine whether the tax effort decreased or increased within the study period.

A summary of the descriptive statistics of the variables used in the model is shown in Table 2.

**Table 2.** Summary of the statistics of the variables used in the model.

| Variable | Unit/Range | Arithmetic Mean | Standard Deviation | Extreme Values | |
| --- | --- | --- | --- | --- | --- |
| | | | | Minimum | Minimum |
| R_PIB | % del PIB | 4.20541 | 1.21971 | 2.26546 | 6.84701 |
| PIBpc | GDP (constant 2011 USD) | 12,689.10 | 5250.76 | 4778.72 | 22,331.23 |
| C_PIB | % of PIB | 54.8901137 | 19.9516675 | 26.9274299 | 105.0440641 |
| Gini | 0–100 | 49.0513889 | 3.4076547 | 43.2 | 56.7 |
| E_sum_pib | % of PIB | 37.47328 | 15.60411 | 12.64000 | 61.77000 |
| Edu_PIB | % of PIB | 3.8975423 | 0.9708080 | 2.3256992 | 6.2673162 |
| IHH | 0–1 | 0.3992382 | 0.2276815 | 0.0820689 | 0.7476185 |
| Min_pib | % of PIB | 4.751 | 5.919 | 0 | 20.917 |
| Oil_pib | % of PIB | 3.460 | 4.700 | 0 | 18.477 |
| Stat | 0–100 | 81.20371 | 9.11059 | 66.66667 | 98.88889 |
| GP_PIB | % of PIB | 29.9565 | 9.5988 | 17.1190 | 54.8 |
| Yeardum | annual | 6.5 | 3.4762778 | 1 | 12 |

Source: Appendix A.

The average customs revenue of the sample countries is 4.21% of the GDP. The countries with the highest average revenue are Bolivia (5.85% of the GDP), Chile (5.34% of the GDP), and Peru (4.38% of the GDP), while Ecuador (3.98% of the GDP), Panama (3.12% of the GDP), and Colombia (2.63% of the GDP) are the countries with the lowest average revenue. As for the Gini index, the country with the highest level of inequality is Colombia (53), closely followed by Panama (52). Bolivia, Chile, Ecuador, and Peru present an average value equal to 48.7, 47.9, 47.6, and 45.6, respectively. The shadow economy as a proportion of GDP ranges from 13.7% to 58.1% of the GDP, with the lowest value for Chile and the highest value for Panama. The rest of the countries have the following average values: Bolivia, 51.5% of the GDP; Peru, 43.5% of the GDP; Ecuador, 30.2% of the GDP; and Colombia, 27.8% of the GDP. The average value for all countries is 37.5% of the GDP.

Another variable considered in the model is government expenditure on education as a proportion of the GDP. On average, Bolivia (5.5% of the GDP), Chile (4.1% of the GDP), and Ecuador (4% of the GDP) have the highest indexes in the ranking, followed by Panama (3.4% of the GDP), Colombia (3.2% of the GDP), and Peru (3.1% of the GDP). For the total sample, the average is equal to 3.9% of the GDP.

In relation to the variable of interest in this paper, the normalized Herfindahl–Hirschman index, the average of the group of countries is 0.4. From a higher to a lower degree of tax complexity, the following stand out on average: Panama (0.13), Colombia (0.21), Ecuador (0.28), Bolivia (0.41), Peru (0.65), and Chile (0.72). Ecuador (11.6% of the GDP), Colombia (4.5% of the GDP), and Bolivia (3.8% of the GDP) are the countries with the highest levels of oil rents. On the other hand, Chile (15.5% of the GDP) and Peru (8.5% of the GDP) have the highest levels for mineral rents, followed by Bolivia with a value equal to 3.4% of the GDP on average. The statistical capacity index presents a better average performance for Chile (94.1), while Bolivia has the lowest performance (70.1). Finally, the average public expenditure of the sample group is equal to 30% of the GDP. The highest average public expenditure was obtained for Bolivia (46.3% of the GDP) and Ecuador (35.9% of the GDP), followed by Colombia (29.6% of the GDP), Panama (23.7% of the GDP), and Peru (22.9% of the GDP). The lowest average public expenditure was obtained for Chile (21.3% of the GDP).

## 5. Results and Discussion

The estimation of the stochastic tax frontier model and the specification of the variables that affect inefficiency were performed using the maximum likelihood procedure, whose model is supported by the qualitative and limited dependent variable model (QLIM) of the SAS/STAT® software. The software estimates the tax effort ($\mu_{it}$) using the method suggested by Battese and Coelli (1988) ("TE1") as well as the one suggested by Jondrow et al. (1982) ("TE2").

The estimation of the stochastic frontier parameters is shown in Table 3. One can see in Table 4 that the model parameters ($\beta$, $\delta$), the random error component ($v_{it}$), and the tax effort component ($\mu_{it}$) have a high degree of significance.

**Table 3.** Estimating the stochastic tax frontier model by means of the maximum likelihood method.

| Variable | Mean | Standard Error | Type |
|---|---|---|---|
| **Ln_R_PIB** | **1.392752** | **0.302494** | **Frontier (Prod) Half-Normal** |
| **Model Fit Summary** | | | |
| Number of Endogenous Variables | | | 1 |
| Endogenous Variable | | | Ln_R_PIB |
| Number of Observations | | | 72 |
| Log Likelihood | | | 57.48817 |
| Maximum Absolute Gradient | | | 0.00549 |
| Number of Iterations | | | 32 |
| Optimization Method | | | Quasi-Newton |
| AIC | | | −96.97634 |
| Schwarz Criterion | | | −76.48635 |
| Sigma | | | 0.16979 |
| Lambda | | | 2.73209 |

| **Parameter Estimates** | | | | | |
|---|---|---|---|---|---|
| Parameter | DF | Estimate | Standard Error | t Value | Approx.Pr > \|t\| |
| **Dependent variable Ln_R_PIB, assuming a half-normal distribution for $\mu_{it}$** | | | | | |
| Intercept | 1 | 56.220266 | 8.878678 | 6.33 | <0.0001 |
| Ln_PIBpp | 1 | −9.447874 | 1.861098 | −5.08 | <0.0001 |
| Ln_PIBpp2 | 1 | 0.473165 | 0.100365 | 4.71 | <0.0001 |
| Ln_c_pib | 1 | 0.557979 | 0.048773 | 11.44 | <0.0001 |
| Ln_gini | 1 | −2.033298 | 0.175228 | −11.60 | <0.0001 |
| Ln_e_sum_pib | 1 | −0.447409 | 0.042602 | −10.50 | <0.0001 |
| Ln_edu_PIB | 1 | −0.394444 | 0.083245 | −4.74 | <0.0001 |
| _Sigma_v | 1 | 0.058359 | 0.020406 | 2.86 | 0.0042 |
| _Sigma_u | 1 | 0.159442 | 0.031864 | 5.00 | <0.0001 |

Source: the estimation.

In relation to the summary of the model's adjustment reported in Table 2, it can be seen that the lambda is much greater than 1 ($\lambda = \mu_{it}/v_{it} = 2.73209$). If the effect of technical inefficiency in the model is calculated $\gamma = \sigma_\mu^2/(\sigma_v^2 + \sigma_\mu^2)$, $\gamma = 0.881856907$, and so 88% of the

discrepancies between the potential and the actual revenue collection are the result of the tax effort. Therefore, judging by the coefficients λ and γ, the model presents an inefficiency effect in the customs revenue collection function. So, it may be estimated using a stochastic frontier model, rejecting the null hypothesis (Ho) that λ = 0.

**Table 4.** Tax effort estimated by truncated regression.

| Summary Statistics of Continuous Response | | | | | |
|---|---|---|---|---|---|
| **Variable** | **Mean** | **Standard Error** | **Type** | **Lower Limit** | **Upper Limit** |
| TE1 | 0.883614 | 0.070368 | Truncated | 0 | 1 |
| **Model Fit Summary** | | | | | |
| Number of Endogenous Variables | | | | 1 | |
| Endogenous Variable | | | | TE1 | |
| Number of Observations | | | | 72 | |
| Log Likelihood | | | | 118.47527 | |
| Maximum Absolute Gradient | | | | 0.0000243 | |
| Number of Iterations | | | | 14 | |
| Optimization Method | | | | Quasi-Newton | |
| AIC | | | | −220.95055 | |
| Schwarz Criterion | | | | −202.73722 | |
| **Parameter Estimates** | | | | | |
| **Dependent variable tax effort, assuming a truncated distribution for $\mu_{it}$** | | | | | |
| Intercept | 1 | 0.648964 | 0.119214 | 5.44 | <0.0001 |
| IHH | 1 | 0.137516 | 0.073344 | 1.87 | 0.0608 |
| Yeardum | 1 | −0.013865 | 0.002727 | −5.09 | <0.0001 |
| Min_pib | 1 | −0.007127 | 0.00329 | −2.17 | 0.0303 |
| Oil_pib | 1 | −0.012981 | 0.00216 | −6.01 | <0.0001 |
| Stat | 1 | 0.003632 | 0.001333 | 2.72 | 0.0065 |
| Gp_pib | 1 | 0.00203 | 0.001155 | 1.76 | 0.0789 |
| _Sigma | 1 | 0.054205 | 0.005719 | 9.48 | <0.0001 |

Source: the estimation.

The model did not achieve convergence when the variable 'Yeardum' was used to determine whether or not the tax effort changed over time according to the half-normal distribution assumed for the term $\mu_{it}$. Therefore, it cannot be guaranteed that there was an improvement due to technological changes during the study period.

Table 5 shows the estimated tax effort for the Customs Administrations of the sample countries.

Tables 6 and 7 give correlation coefficient values between the variables of this model. The analysis indicates that the correlation effects between the variables are not strong, except for the variable related to mineral rents and the tax complexity index. However, we preferred to keep the mineral rents variable because its contribution is representative of the economy of many countries in the sample.

On the other hand, Tables 8 and 9 show the residual normality test and a good fit for the residues, so the null hypothesis that the residues have a normal distribution cannot be rejected.

According to the results of the parameter estimation shown in Table 3, we can make the following points.

The convex relationship between GDP per capita and tax revenue as a percentage of GDP can be interpreted as follows: The growth rate of GDP per capita is not sufficient to produce increases in customs revenue. However, the relationship between GDP per capita squared and tax revenue as a percentage of GDP is positive, which means that if the GDP per capita increases at a good rate, the tax revenue will also increase.

The parameters related to the level of trade openness, the Gini index, and the shadow economy show the expected signs. Ceteris paribus, an average increase of 1% in trade

openness would improve customs revenue by 0.6%. On the other hand, a 1% increase in the economic inequality index or in the level of the shadow economy would reduce tax revenues by 2% and 0.4%, respectively. Regarding the variable that measures public spending on education as a percentage of GDP, a decreasing impact on customs revenue was observed (−0.4%), confirming the ambiguity identified by Cyan et al. (2013) regarding the effect of educational level on revenues, such that knowledge could be applied to evade taxes.

The simplification of customs duties and taxes, measured by the normalized HHI index, positively and significantly affected the tax effort, i.e., the simplification of the customs tax system with respect to the tax effort is in a 1 to 0.14 ratio. In the case of Ecuador, for example, on average, if the tax simplification increased from 0.28 to 0.38, the tax effort would increase from 0.81 to 0.824.

The negative sign of the estimator of the variable Yeardum indicates that the tax effort decreased over time (2006–2017). The statistical capacity index, which was used as a proxy for the level of accountability and transparency of the government, shows the expected sign, i.e., the tax effort tends to improve when citizens have statistical information of sufficient quality to allow them to question the actions of the government. To illustrate the event, an improvement of one unit of the statistical capacity index in Ecuador (81.2 to 82.2) would mean an increase in the tax effort by 0.004 points (0.81 to 0.814) in average terms. On the other hand, an increase of 1 point in mineral and oil rents as a percentage of GDP, ceteris paribus, would deteriorate the tax effort by 0.007 and 0.013 points, respectively. This result supports the conjecture that high public revenues from natural resources such as minerals and especially oil relax tax controls.

In this regard, taking the Ecuadorian case as a reference, in Figure 1 we observe a decrease in the tax effort that coincides with the years when the price of a barrel of oil (WTI) increased. On the other hand, it can be seen that the country's total debt began to increase from 2009 onwards, with this increase not being very significant during the period 2009–2014 when oil prices had an upward trend and the tax effort gradually decreased.

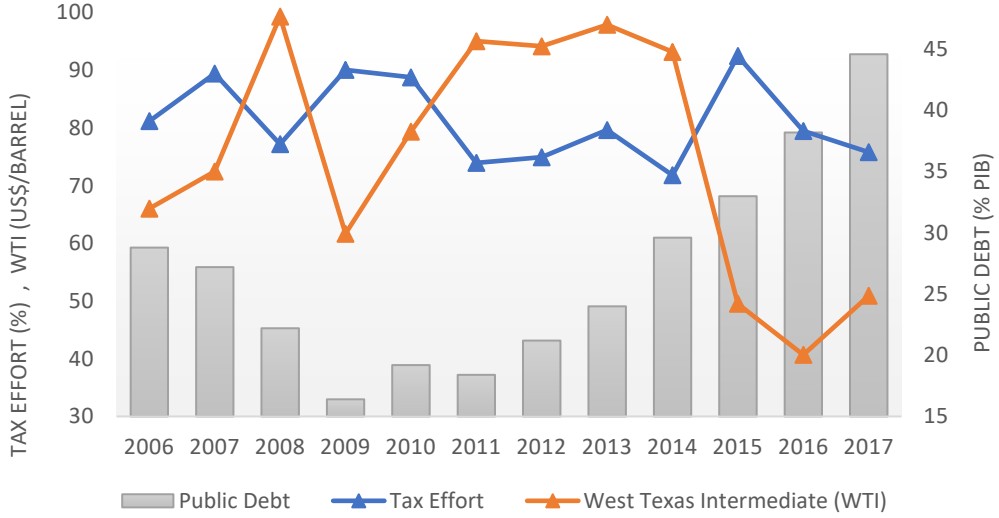

**Figure 1.** Periodic comparison of Ecuador's total public debt (% GDP), WTI oil price (USD/barrel), and tax effort (%). Source: Appendix A.

**Table 5.** Tax effort score by country and year.

| Country | Bolivia | | Chile | | Colombia | | Ecuador | | Panama | | Peru | |
|---|---|---|---|---|---|---|---|---|---|---|---|---|
| Year | $TE_1$ | $TE_2$ | $TE_1$ | $TE_2$ | $TE_1$ | $TE_2$ | $TE_1$ | $TE_2$ | $TE_1$ | $TE_2$ | $TE_1$ | $TE_2$ |
| 2006 | 0.92229 | 0.92130 | 0.82023 | 0.81900 | 0.95765 | 0.95713 | 0.81178 | 0.81056 | 0.89813 | 0.89697 | 0.95589 | 0.95534 |
| 2007 | 0.92997 | 0.92906 | 0.88098 | 0.87975 | 0.95739 | 0.95686 | 0.89420 | 0.89302 | 0.93254 | 0.93166 | 0.96334 | 0.96291 |
| 2008 | 0.84175 | 0.84049 | 0.97212 | 0.97183 | 0.95420 | 0.95363 | 0.77177 | 0.77061 | 0.95310 | 0.95251 | 0.97104 | 0.97074 |
| 2009 | 0.85336 | 0.85210 | 0.93443 | 0.93358 | 0.91821 | 0.91718 | 0.90055 | 0.89940 | 0.89809 | 0.89692 | 0.93215 | 0.93127 |
| 2010 | 0.87608 | 0.87483 | 0.94575 | 0.94505 | 0.95857 | 0.95807 | 0.88759 | 0.88638 | 0.94393 | 0.94320 | 0.91729 | 0.91625 |
| 2011 | 0.77658 | 0.77541 | 0.91308 | 0.91201 | 0.86756 | 0.86631 | 0.73935 | 0.73824 | 0.91282 | 0.91175 | 0.83511 | 0.83387 |
| 2012 | 0.78662 | 0.78544 | 0.93900 | 0.93821 | 0.81844 | 0.81722 | 0.74912 | 0.74800 | 0.89051 | 0.88931 | 0.87856 | 0.87733 |
| 2013 | 0.87873 | 0.87749 | 0.94077 | 0.93999 | 0.77392 | 0.77275 | 0.79639 | 0.79520 | 0.87244 | 0.87119 | 0.93274 | 0.93187 |
| 2014 | 0.95511 | 0.95455 | 0.92857 | 0.92765 | 0.79335 | 0.79216 | 0.71800 | 0.71692 | 0.83401 | 0.83277 | 0.95085 | 0.95022 |
| 2015 | 0.97886 | 0.97868 | 0.95031 | 0.94967 | 0.81109 | 0.80987 | 0.92473 | 0.92377 | 0.89501 | 0.89383 | 0.96063 | 0.96016 |
| 2016 | 0.91535 | 0.91430 | 0.92759 | 0.92665 | 0.75642 | 0.75528 | 0.79470 | 0.79351 | 0.86754 | 0.86629 | 0.95085 | 0.95023 |
| 2017 | 0.90063 | 0.89948 | 0.91041 | 0.90931 | 0.73180 | 0.73070 | 0.75795 | 0.75681 | 0.81544 | 0.81422 | 0.93490 | 0.93405 |
| Mean | 0.88461 | 0.88359 | 0.92194 | 0.92106 | 0.85822 | 0.85726 | 0.81218 | 0.81104 | 0.89280 | 0.89172 | 0.93195 | 0.93119 |

Note: "TE1" and "TE2" are the methods for the estimation of technical efficiency proposed by Battese and Coelli (1988) and Jondrow et al. (1982), respectively.

**Table 6.** Correlation matrix of the variables used to determine the potential capacity.

| | Ln_PIBpp | Ln_PIBpp2 | Ln_C_PIB | Ln_Gini | Ln_E_sum_pib | Ln_Edu_PIB |
|---|---|---|---|---|---|---|
| | Pearson Correlation Coefficient, N = 72 | | | | | |
| | Prob > \|r\| Assuming H$_0$: Rho = 0 | | | | | |
| Ln_PIBpp | 1 | | | | | |
| Ln_PIBpp2 | 0.99967 (<0.0001) | 1 | | | | |
| Ln_C_PIB | 0.11184 (0.3496) | 0.12283 (0.304) | 1 | | | |
| Ln_Gini | 0.00814 (0.9459) | 0.01018 (0.9324) | 0.07302 (0.5421) | 1 | | |
| Ln_E_sum_pib | −0.50701 (<0.0001) | −0.50768 (<0.0001) | 0.31014 (0.008) | 0.16473 (0.1667) | 1 | |
| Ln_Edu_PIB | −0.4005 (0.0005) | −0.38583 (0.0008) | 0.18416 (0.1215 | −0.21645 (0.0678) | 0.00286 (0.981) | 1 |

Source: the estimation.

**Table 7.** Correlation matrix of the variables used to determine the tax effort.

| | IHH | Min_pib | Oil_pib | Stat | Gp_pib |
|---|---|---|---|---|---|
| | Pearson Correlation Coefficient, N = 72 | | | | |
| | Prob > \|r\| Assuming H$_0$: Rho = 0 | | | | |
| IHH | 1 | | | | |
| Min_pib | 0.84589 (<0.0001) | 1 | | | |
| Oil_pib | −0.25482 (0.0308) | −0.43442 (0.0001) | 1 | | |
| Stat | 0.57777 (<0.0001) | 0.59179 (<0.0001) | −0.21168 (0.0743) | 1 | |
| Gp_pib | −0.26038 (0.0272) | −0.44105 (0.0001) | 0.37837 (0.001) | −0.63918 (<0.0001) | 1 |

Source: the estimation.

**Table 8.** Test of residual normality for the stochastic tax frontier model.

| Variable: Resid_Ln_R_PIB (Ln_R_PIB Residual) | | | | |
|---|---|---|---|---|
| Test | Statistical | | *p* Value | |
| Shapiro–Wilk | W | 0.971471 | Pr < W | 0.1009 |
| Kolmogorov–Smirnov | D | 0.082440 | Pr > D | >0.1500 |

**Table 9.** Normality test for the model of tax-effort-determining variables.

| Variable: Resid_TE1 (TE1 Residual) | | | | |
|---|---|---|---|---|
| Test | Statistical | | *p* Value | |
| Shapiro–Wilk | W | 0.973903 | Pr < W | 0.1396 |
| Kolmogorov–Smirnov | D | 0.094733 | Pr > D | >0.1075 |

The sign of the parameter related to public expenditure as a proportion of GDP is positive. This is evidence that the level of expenditure made by the public sector contributes to improve the tax effort. This could be explained by the fact that providing improved public goods and services motivates citizens to improve their tax compliance since they perceive the payment of their taxes to be reversed in their own interest.

Figure 2 shows the average tax effort for the countries in the study. It should remember that the average tax effort is the difference between potential and actual revenues. The average of the sample was 88.3%. During the study period, Peru was the country that was closest to its potential capacity, while Ecuador occupied the last position.

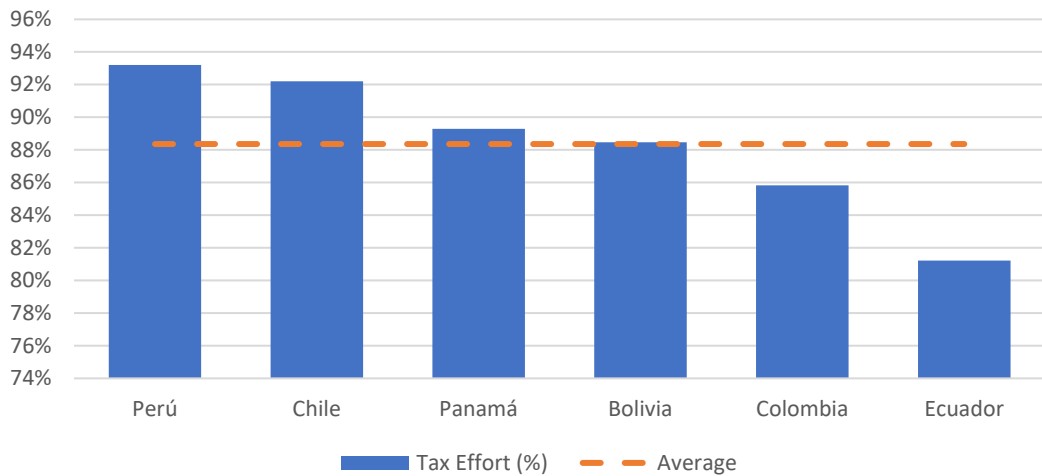

**Figure 2.** Tax effort (overall country average, 2006–2017). Source: Appendix A.

Figure 3 shows a comparison between the customs revenue collection as a proportion of the value of FOB imports and tax efforts. Countries such as Ecuador and Colombia have a lower than average level of customs revenue collection because their tax effort is relatively low. Meanwhile, Panama has a low level of customs revenue collection for its level of tax effort because its potential capacity is low, presumably due to the large size of the shadow economy (58% on average) compared with the other countries in the sample.

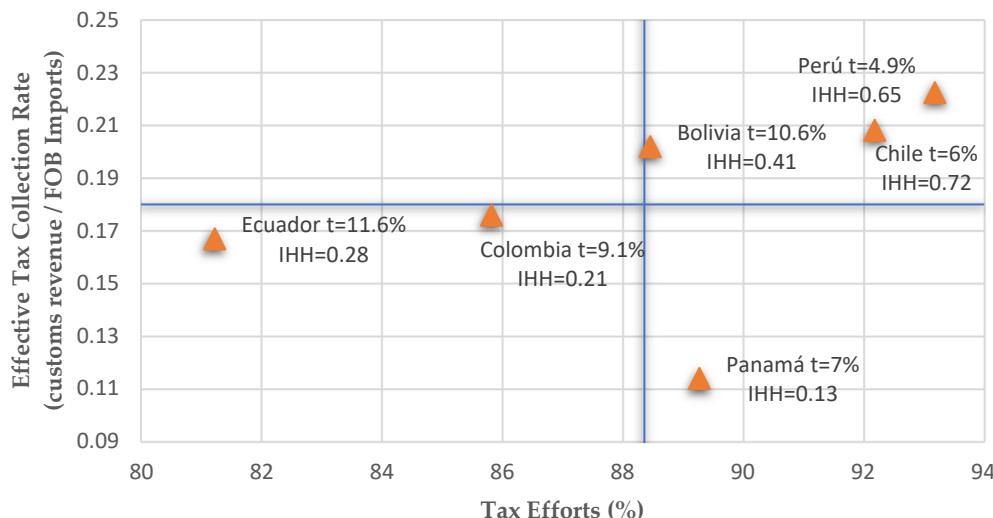

**Figure 3.** Revenue collected by Ecuador's Customs Administration (% of FOB imports) versus tax effort (%): average for the period 2006–2017.

Bolivia is at the limit of the sample average in terms of its tax effort. Its revenue collection maintains a good rate, driven by the growth in imports. Additionally, one can see that it presents a competitive advantage with respect to the degree of tax complexity, which is much lower than that of Ecuador and Colombia.

Chile and Peru are the countries that show the best performance in terms of the degree of revenue collection and tax effort. They are also the countries with the lowest average MFN tariff rate and the highest degree of tax simplicity.

This fact is consistent with what the literature indicates: Having a variety of product codes and nomenclatures, exceptions, and different types and levels of tariffs provides opportunities to avoid paying taxes. Therefore, it is much easier for the Customs Administration to control a tax system with the lowest possible complexity.

## 6. Conclusions

In order to analyze whether Ecuador's customs tax system presents any degree of complexity that affects tax payment compliance, a stochastic tax frontier function was modeled for the period from 2006 to 2017 with a sample composed of six countries (Bolivia, Chile, Colombia, Ecuador, Panama, and Peru) using the methodology proposed by Battese and Coelli (1995). The method used to estimate the parameters of this model is the maximum likelihood (MLE) method.

The following independent variables were selected to specify the model for determining potential revenue: GDP per capita by purchasing power parity (constant 2011 international dollarrs), merchandise trade (% of GDP), the Gini index, public spending on education (% of GDP) (Pessino and Fenochietto 2010), and the shadow economy (% of GDP) (Schneider 2012; Schneider and Medina 2018).

As for the model of the determinants that explain the tax effort, the tax complexity index measured by the normalized Hirschman–Herfindahl index was used, whose calculation was based on the statistics of customs revenue collection by tax type. Likewise, the mineral revenue (% of GDP) and oil revenue (% of GDP), the statistical capacity index as a measure of the degree of transparency and accountability, public expenditure (% of GDP), and a dummy variable were used to determine the increasing or decreasing trend of the tax effort through the time. The major findings of this publication can be summarized as follows:

- The countries whose customs tax system has a lower degree of complexity have a better level of collection and tax effort (Peru, Chile, and Bolivia). Therefore, the greatest possible degree of simplicity in the design of the tax system is desirable.
- Panama's potential customs revenue collection is very close to its actual customs revenue collection, but its actual revenue collection is low compared with the other countries in the sample. Therefore, work could be carried out on aspects that help to reduce the shadow economy in order to improve tax revenues.
- An improvement in the quality and dissemination of statistical data helps to improve transparency and the possibility of questioning government decision-making, which contributes to improving the tax effort. Notably, labor efficiency levels tend to increase when there are means to audit through tangible results.
- When revenue levels from non-renewable natural resources such as minerals and oil increase, the tax effort shows a downward trend. In the case of Ecuador, a certain degree of laziness in tax control was observed during the years when the WTI crude oil price increased that did not justify the increase in the level of debt. In contrast, a greater tax effort could have been made to increase customs revenues, for example through a reduction in the complexities of the customs tax system.

**Author Contributions:** Conceptualization, A.D.V.; formal analysis, J.G.A.; investigation, J.G.A. and A.D.V.; methodology, J.G.A.; resources, A.D.V.; writing—review and editing, J.G.A. and A.D.V. All authors have read and agreed to the published version of the manuscript.

**Funding:** This research received no external funding.

**Institutional Review Board Statement:** Not applicable.

**Informed Consent Statement:** Not applicable.

**Data Availability Statement:** Publicly available datasets were analyzed in this study. This data can be found here: https://www.aduana.gob.bo/aduana7/content/bolet%C3%ADn-de-recaudaciones-0 (on 21 October 2019); https://www.aduana.cl/aduana/site/edic/base/port/estadisticas.html (accessed on 21 October 2019); https://www.dian.gov.co/dian/cifras/Paginas/EstadisticasRecaudo.aspx (accessed on 21 October 2019); https://www.ine.gob.bo/index.php/prod-interno-bruto-anual/introduccion-3 (accessed on 21 October 2019); https://si3.bcentral.cl/Siete/secure/cuadros/home.aspx (accessed on 21 October 2019); http://www.banrep.gov.co/es/estadisticas/producto-interno-bruto-pib (accessed on 21 October 2019); https://www.aduana.gob.ec/novedades/recaudaciones (accessed on 21 October 2019); https://www.ana.gob.pa/w_ana/index.php/qui

enes-somos/cifras-y-gestiones/recaudaciones (accessed on 21 October 2019); http://www.sunat.gob.pe/estadisticasestudios/ingresos-recaudados.html (accessed on 21 October 2019); https://contenido.bce.fin.ec/home1/estadisticas/bolmensual/IEMensual.jsp (accessed on 21 October 2019); https://www.contraloria.gob.pa/inec/ (accessed on 21 October 2019); http://www.bcrp.gob.pe (accessed on 21 October 2019); http://data.un.org (accessed on 21 October 2019); https://datos.bancomundial.org (accessed on 21 October 2019); https://estadisticas.cepal.org (accessed on 21 October 2019); https://www.imf.org/en/Publications/WP/Issues/2018/01/25/Shadow-Economies-Around-the-World-What-Did-We-Learn-Over-the-Last-20-Years-45583 (accessed on 21 October 2019); http://ftp.iza.org/dp6423.pdf (accessed on 21 October 2019); https://www.imf.org/external/datamapper/DEBT1@DEBT/OEMDC/ADVEC/WEOWORLD (accessed on 21 October 2019).

**Conflicts of Interest:** The authors declare no conflict of interest.

## Appendix A

**Table A1.** Data source for variables used to estimate the potential capacity and determinants of tax effort: stochastic frontier model.

| Variable | Data Sources Available at: |
|---|---|
| Customs Revenue Collection | https://www.aduana.gob.bo/aduana7/content/bolet%C3%ADn-de-recaudaciones-0 (accessed on 21 October 2019)<br>https://www.aduana.cl/aduana/site/edic/base/port/estadisticas.html (accessed on 21 October 2019)<br>https://www.dian.gov.co/dian/cifras/Paginas/EstadisticasRecaudo.aspx (accessed on 21 October 2019)<br>https://www.ine.gob.bo/index.php/prod-interno-bruto-anual/introduccion-3 (accessed on 21 October 2019)<br>https://si3.bcentral.cl/Siete/secure/cuadros/home.aspx (accessed on 21 October 2019)<br>http://www.banrep.gov.co/es/estadisticas/producto-interno-bruto-pib (accessed on 21 October 2019) |
| Current GDP and FOB Imports | https://www.aduana.gob.ec/novedades/recaudaciones/ (accessed on 21 October 2019)<br>https://www.ana.gob.pa/w_ana/index.php/quienes-somos/cifras-y-gestiones/recaudaciones (accessed on 21 October 2019)<br>http://www.sunat.gob.pe/estadisticasestudios/ingresos-recaudados.html (accessed on 21 October 2019)<br>https://contenido.bce.fin.ec/home1/estadisticas/bolmensual/IEMensual.jsp (accessed on 21 October 2019)<br>https://www.contraloria.gob.pa/inec/ (accessed on 21 October 2019)<br>http://www.bcrp.gob.pe/ (accessed on 21 October 2019) |
| GDP per capita, PPP (constant 2011 international dollars) | http://data.un.org/ (accessed on 21 October 2019) |
| Gini; Oil and Mineral Rents (GDP %); Statistical Capacity; exchange rate; WTI Crude; | https://datos.bancomundial.org/ (accessed on 21 October 2019) |
| Public Education Expenditure (GDP %) | https://estadisticas.cepal.org (accessed on 21 October 2019) |
| Shadow Economy (GDP %) | https://www.imf.org/en/Publications/WP/Issues/2018/01/25/Shadow-Economies-Around-the-World-What-Did-We-Learn-Over-the-Last-20-Years-45583 (accessed on 21 October 2019)<br>http://ftp.iza.org/dp6423.pdf (accessed on 21 October 2019) |
| Public Expenditure (GDP %) | https://www.imf.org/external/datamapper/DEBT1@DEBT/OEMDC/ADVEC/WEOWORLD (accessed on 21 October 2019) |

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
