# Peer review of "Effect of the Complexity of the Customs Tax System on the Tax Effort"

_economies, doi:10.3390/economies10030055_

Round 1
Reviewer 1 Report
The authors have addressed the comments appropriately.
Author Response
We thank the reviewer #1 for the constructive and insightful comments, which have helped us to substantially improve our manuscript.
Reviewer 2 Report
Effect of the Complexity of the Customs Tax System on the Tax Effort
In my first review, I stated that the definitions in the paper were expressed in a confusing way. In the revised version of the paper, the authors have made some effort to express the ideas more clearly but there is still a tendency to use several different expressions to mean apparently the same thing.
I understand the authors to be saying the following:
#1 there is a maximal level of tax revenue – this is what they call tax capacity. They explain this tax capacity by constructing a tax production frontier using variables labelled x and including a random error, v. This is equation 1.
#2 actual revenue falls short of the maximal tax capacity. This is explained by subtracting a nonnegative inefficiency term, u. Therefore, they have a stochastic frontier analysis of the tax capacity frontier.
#3 the inefficiency term, u, is determined by a further set of variables, Z, in a deterministic technical efficiency effects relationship as suggested by Battese and Coelli (1995) and others. This is equation 2.
The technical efficiency of the tax administration is the usual measure: TE = exp(-u). This is equation 3. The authors insist on calling this standard measure of technical efficiency by the name ‘tax effort’. But this is just confusing. It implies that high effort is the same as high technical efficiency whereas most of the production economics literature associates high effort with excessive use of inputs, i.e., low efficiency. This was a major source of confusion to me in my review of the first version of the paper. I cannot understand why the authors insist on this confusing definition. It surely would be much simpler to state that they wish to measure the relative technical efficiency of the different tax systems, an avoid confusing terms like tax effort which can have different interpretations.
This is all that the paper needs in sections 2-3. With these clarifications, the paper made more sense to me.
Section 4 gives the important specifications of equations 1 and 2. Clearly the success of the paper depends on a careful choice of variables in this context. The authors have made some interesting suggestions for relevant variables, but they have not suggested any theoretical justification for the choices made. Superficially they seem plausible, but the absence of a theoretical model of tax efficiency is noticeable. The specifications therefore have to be defended as coming from the cited literature. Clearly the size of the shadow economy is critical but the authors have simply extrapolated these data from other sources. In the inefficiency equation the specification of the IHH variable is useful and the idea of the statistical capacity of the national administration is interesting.
The description of the estimation methodology is confusing. The authors refer to Battese and Coelli (1988) and Jondrow et al (1982) for the construction of the technical efficiency terms. This is correct: these papers give two different procedures for calculating the conditional mean inefficiency component. However the technical efficiency effects model suggested by Battese and Coelli (1995) which the authors described in sections 2-3 is different. It estimates their equations 2 and 3 in a single step (remembering that equation 3 is a deterministic component of equation 2 not an additional econometric equation). This needs to be clarified in the paper.
Overall, the results are interesting and persuasive. I understand what the authors have done in this paper more clearly in the revised version. They still have a habit of using several expressions to refer to a single term. I advise them to stick to the term efficiency of the tax collection system instead of using the phrase tax effort.
Reviewer 3 Report
The issue raised by the authors is indeed an important one, and is interesting from both scientific and practical perspectives. The abstract accurately conveys the content of the paper. The introduction should be abbreviated, and it should be connected with the paper aim. I would like to mention that the authors have not comprehensively studied the literature on the issue published over the last five years. References are correct. The conclusion is consistent with presented arguments and evidence. The results complete previous results on the matter and are supported by references.
The conclusions should be abbreviated, and it should be connected with the paper aim. It is not necessary to list the explanatory variables selected for the model. There should several important scientific results having novelty, future and application in the considered context for the country.
Therefore, the article needs to be revised and corrected.
Author Response
We want to thank you for your comments and revision recommendations. We appreciate very much your suggestions.
Respect literature sources, we have selected those which connected with the aim of the research. Of course, there are issues published in the last years related to our research. But most of them are based on models or ideas from other research we cited in our paper.
We cannot cite all the published papers and research related to our paper. We just take the most significant number of papers that give us support to our ideas or are supposed to be an important advance in knowledge.
We make changes in the introduction and the conclusions following reviewer’s recommendations.
This manuscript is a resubmission of an earlier submission. The following is a list of the peer review reports and author responses from that submission.
Round 1
Reviewer 1 Report
Journal: Economies (ISSN 2227-7099)
Manuscript ID: economies-1501980
Manuscript Type: Research Paper
Title: Effect of the Complexity of the Customs Tax System on the Tax Effort: evidence from Ecuador
Review Report
Using a stochastic production frontier model, this research empirically studies the effects of the tax complexity and other tax-related factors on the efficiency of Ecuadorian Customs Administration compared to the other five neighboring countries – Bolivia, Chile, Columbia, Peru, and Panama. It documents that 1) the countries with a lower degree of complexity of the customs tax system have a better level of collection and tax effort (Peru, Chile, and Bolivia); 2) only Panama’s potential customs revenue collection is very close to its actual customs revenue collection; 3) an improvement in the quality and dissemination of statistical data helps improve transparency and the possibility of questioning government decision making; 4) when revenue levels from non-renewable natural resources such as mineral and oil increase, the tax effort show a downward trend. It is a reasonable research project and I will go straight to my points.
- Defining the key terms as early as possible: For readers, please define the key terms as early as possible. For example, the term tax effort can be defined in the abstract to help readers understand what research questions are in this study. Once defined, please do not deviate from it. That said, I don’t understand what it means by “In the SFA methodology, the tax effort, which by definition is the ratio of total revenue as a share of GDP to a country's per capita income, is measured by relating effective revenue to potential (economic) revenue (refer to page 3 of the manuscript).” Please redefine all key terms in the study. If needed, please consider adding an Appendix showing all key terms.
- Explain the implication of the key measurement terms: Let’s take a look at the same term, tax effort. Authors define tax effort as the ratio between actual revenue and fiscal capacity (refer to page 2). So, if actual revenue is $1 M and fiscal capacity is $2 M, then the tax effort would be 0.5. If actual revenue is $1 M and fiscal capacity is $10 M, then the tax effort would be 0.1. So, higher the tax effort better ability of the tax administration to collect maximum tax revenues. I think authors may need to switch this term from Uit to (1-Uit) (refer to page 4, Equation 2). As of now, there is a discrepancy between what this proxy measures and what authors are trying to understand/explain. If I am missing something here, please explain it fully.
- Econometric issues: In this study, there is no clearly established causal relationship between the degree of complexity in the customs tax system and tax revenues collected, tax effort, and dissemination of national statistical data. The reported results are largely driven by the author's research design – not based on any theoretical argument and/or prior findings in the literature. Authors may consider conducting before-and-after tests if it is available.
- Professional English editing and proofreading may improve the quality of writing significantly and authors may consider taking advantage of that.
Again, it is an interesting research project and I wish you the very best of luck in publishing this research project.
Regards,
Reviewer 2 Report
This manuscript analyzes the effects of the complexity of the customs tax system on the tax effort, defined as the ratio between actual customs revenue and fiscal capacity, using a panel data of six Latin American countries (Bolivia, Chile, Colombia, Ecuador, Peru and Panama). Following Battese and Coelli (1995) and Pessino and Fenochietto (2013) among others, an econometric model of the stochastic production frontier was used and adapted to construct a stochastic fiscal frontier.
Overall, the topic is relevant and the manuscript derives meaningful results from a policy perspective for developing countries.
Below are some comments and suggestions that I would like to make:
- The title “evidence from Ecuador” is not appropriate. The paper uses a panel data of six Latin American countries, and is reporting the results for the six countries.
- Related to the above point, the paper needs overall adjustments in writing and motivation: the paper is not providing evidence only for Ecuador. Please adjust relevant parts of the paper so that the writing is consistent with what has been done: for example, line 9 and 569; in particular, the part from line 33 to line 59 should be reviewed in order to give some general motivation for developing countries, and not only for Ecuador.
- The part from line 205 to line 244 can be summarized using tables.
- The part from line 245 to line 258 should be eliminated (an error of repetition).
- The title “3.2 Data Source” is not appropriate. The following content is describing the model.
- In tables, points and commas are mixed for decimal numbers.
Reviewer 3 Report
Effect of the Complexity of the Customs Tax System on the Tax Effort: evidence from Ecuador
- The paper uses stochastic frontier analysis to analyse the stochastic fiscal frontier in a panel data sample of countries in South America from 2006-17.
- The introduction is useful in motivating what looks like an interesting problem.
- The literature review begins with a definition of tax effort which is not clear, page 3 lines 120-21: it appears to be (Total Revenue as a share of GDP) divided by (GDP per person). Is this not simply Total revenue per person? Concepts of effective, actual and potential revenue are then mentioned without clear definitions. It seems that effective and actual revenue are the same – why use two different terms?
- The stochastic frontier model suggested is a form of production function in which the dependent variable is ‘actual or effective revenue’. The random error component representing inefficiency is defined to be the tax effort and it is determined by a number of additional explanatory variables using the Battese and Coelli (1995) technical efficiency effects model. The paper should explain why the one-sided inefficiency component is a measure of tax effort. Surely, it is the wrong way round. High Inefficiency should mean low tax effort, but the paper seems to state that the opposite is true. This does not make sense and probably invalidates the whole paper.
- The model specification is in section 3.1. Note that the last part of section 3.1 is a repetition of the preceding part. A range of variables is listed and there is an explanation of each of them in the next section. Each of the explanations seems reasonably well based although there is no explanation of an underlying economic model of fiscal effort, and the list seems to have been drawn up mainly on the basis of possible correlations. The use of the HHI as a measure of complexity is interesting and deserves greater explanation.
- Results are shown in tables 2 and 3 in section 4. It is not clear whether log-mean-corrected data have been used. If so the coefficient values seem strange. There is some commentary on the estimated regression parameters, but it is noticeable that GDP per capita has significant but negative relationship with the share of tax revenue. Why should this be? The explanation below table 8 in terms of a Laffer curve effect is not convincing. It reflects the absence of a theoretical model of tax revenue determination. There is weak evidence of a positive complexity effect using HHI on inefficiency but the t-value is low, table 3. Table 3’s dependent variable is the expected value of inefficiency using the Battese Coelli model. More importantly if lower complexity is positively related to inefficiency as table 3 suggests, then this means that it reduces tax effort. This is opposite to the claim made in the paper.
- The paper contains a major confusion in the model. The confusion is that it equates high expected value of the inefficiency component of the random error with strong tax effort. This makes no sense. The authors need to rethink the way that they have formulated the model or to explain it much more clearly.